# Intracranial Aneurysm Rupture after SARS-CoV2 Infection: Case Report and Review of Literature

**DOI:** 10.3390/pathogens11060617

**Published:** 2022-05-24

**Authors:** Dilaware Khan, Soheil Naderi, Mostafa Ahmadi, Askar Ghorbani, Jan Frederick Cornelius, Daniel Hänggi, Sajjad Muhammad

**Affiliations:** 1Department of Neurosurgery, Medical Faculty, Heinrich-Heine-University, Moorenstrasse 5, 40225 Düsseldorf, Germany; dilaware.khan@med.uni-duesseldorf.de (D.K.); cornelius@med.uni-duesseldorf.de (J.F.C.); daniel.haenggi@med.uni-duesseldorf.de (D.H.); 2Department of Neurosurgery, Imam Khomeini Hospital, Tehran University of Medical Science, Tehran 1419733141, Iran; soheilnaaderi@gmail.com (S.N.); dr.ahmadi.mostafa@gmail.com (M.A.); 3Department of Neurology, Tehran University of Medical Sciences, Tehran 1419733141, Iran; as.ghorbani67@gmail.com; 4Department of Neurosurgery, University Hospital Helsinki, Topeliuksenkatu 5, 00260 Helsinki, Finland

**Keywords:** COVID-19, neurosurgery, neurovascular complications, aSAH

## Abstract

Background: SARS-CoV virus infection results in a dysbalanced and severe inflammatory response with hypercytokinemia and immunodepression. Viral infection triggers systemic inflammation and the virus itself can potentially cause vascular damage, including blood–brain barrier (BBB) disruption and alterations in the coagulation system, which may result in cardiovascular and neurovascular events. Here, we review the literature and present a case of COVID-19 infection leading to an aneurysmal subarachnoid haemorrhage (aSAH). Case Description: A 61-year-old woman presented with dyspnea, cough, and fever. She had a history of hypertension and was overweight with a body mass-index of 34. There was no history of subarachnoid hemorrhage in the family. Due to low oxygen saturation (89%) she was admitted into ICU. A chest CT showed a typical picture of COVID-19 pneumonia. The PCR-based test of an oropharyngeal swab was COVID-19-positive. In addition to oxygen support she was prescribed with favipiravir and hydroxychloroquine. She experienced a sudden headache and lost consciousness on the second day. Computer tomography (CT) with CT-angiography revealed a subarachnoid haemorrhage in the basal cisterns from a ruptured anterior communicating artery aneurysm. The aneurysm was clipped microsurgically through a left-sided standard pterional approach and the patient was admitted again to the intensive care unit for further intensive medical treatment. Post-operatively, the patient showed slight motor dysphasia. No other neurological deficits. Conclusion: Systemic inflammation and ventilator support-associated blood pressure fluctuations may trigger aneurysmal subarachnoid haemorrhage secondary to COVID-19 infection. COVID-19 infection could be considered as one of the possible risk factors leading to instability and rupture of intracranial aneurysm.

## 1. Introduction

The number of SARS-CoV-2 virus infected patients is increasing dramatically and has now reached above 520 million confirmed infected individuals with over 6.2 million deaths worldwide. The World Health Organisation declared the COVID-19 disease as a pandemic in February 2020. The disease started in December 2019 in Wuhan, China, and spread rapidly throughout the entire globe. Intensive epidemiological and biological research led to the identification of novel corona virus 2019 (COVID-19) that can infect both animals and humans [1,2]. The COVID-19 infection is respiratory in nature and can range from a common cold with mild symptoms to severe acute respiratory syndrome (SARS) with respiratory failure [3,4,5], but can also involve other organs and systems including the cardiovascular system, the gastrointestinal tract, and the central nervous system (CNS) [6,7]. The involvement of other organs, especially the cardiovascular system, has been shown to pose significant additional mortality in COVID-19 infected patients [8]. The involvement of the central nervous system during COVID-19 infection, especially vascular damage leading to intracranial bleeding, is being explored. Until now, 14 cases of subarachnoid haemorrhage after COVID-19 infection have been reported in the world literature. Here we review these cases and present a demonstrative case of SAH after COVID-19 infection.

## 2. Methods

We reviewed the current literature with clinical reports of SAH in COVID-19 patients. We searched for the clinical reports using the search keyword ‘Subarachnoid Haemorrhage (SAH)’ in PubMed and Google Scholar. We collected the published data and performed statistical analysis for the available data with respect to demographics, comorbidities, lab reports, and clinical outcomes. In this study, we included the articles that had reported SAH in COVID-19-positive patients. In the literature, there were 14 cases of SAH. Out of which seven cases were non-aneurysmal SAH and in two cases aetiology was not determined, which have not been included in the statistical analysis.

## 3. Results

### 3.1. Clinical Representation of COVID-19-Positive SAH Cases in Published Literature

Neurological manifestations were first reported in a Chinese study, where 36.4% of COVID-19-positive patients showed neurological complications [7]. Including our case, there are total of six cases of aneurysmal SAH in COVID-19-positive patients (with a gender distribution of one man and five women; with an average age of 52 ± 15 years, oldest 68 and youngest 31 years old) have been reported.

In addition to typical COVID-19 symptoms, which include fever, fatigue, dry cough, and shortness of breath [2,4,5], the patients had initial symptoms such as nausea, vomiting, and dyspnea, sometimes in combination with neurological symptoms including anosmia, headache, vomiting, impaired consciousness, and loss of consciousness (Table 1) [9,10,11,12].

One patient (36 years old female) had only systemic arterial hypertension [9], one patient (68 years old female) had a previously detected right posterior communicating artery aneurysm [11], and one patient was overweight with a history of hypertension (Table 1).

One patient needed mechanical ventilation support [9], two patients required oxygen therapy [11], one patient had respiratory insufficiency [10], and one patient had relatively mild respiratory symptoms without progression to acute respiratory distress syndrome [12] Table 1.

The SAH grades were given for all patients; one patient had Federation of Neurological Surgeons (WFNS) four grade for SAH [10], one patient had Hunt and Hess (HH) grade one and Fisher grade two [11], one patient had HH grade two and WFNS grade one [13], one patient had HH grade three [12], and one patient had Fisher grade four [9]. In two patients, the development of hydrocephalus was reported [9,12].

### 3.2. General Treatment

One patient was treated with the antibiotics Ceftriaxone, Meropenem, and Vancomycin [9], and two patients were treated with antiviral Favipiravir and Acyclovir IV [9]. One patient was given the antimalarial drug hydroxychloroquine.

### 3.3. Treatment for SAH

In three patients the aneurysm was clipped microsurgically [10,13], in one patient the aneurysm was occluded with coils [11], in one patient balloon-assisted embolization was performed and the aneurysm was occluded completely [9], and in one patient the aneurysm was treated with a flow-diverter [12]. An external ventricular drain was implanted for three patients to relieve acute hydrocephalus [9,12].

## 4. Outcomes

Four patients were discharged for rehabilitation [9,10,12,14] and two patients were discharged as asymptomatic (mRS = 0) [11,13].

### 4.1. Viral Neurotropism

Viruses from almost all families have been suggested to infect the central nervous system. Many viruses have been reported to infect the nervous system including the polio virus, cytomegalovirus, entero virus, varicella zoster, mumps, toga viruses, herpes simplex virus, rabies virus, and west-Nile viruses. The viral infections of zoonotic origin can be highly virulent and neuro-invasive. As CNS neurons are irreplaceable, when these neurons are infected by zoonotic viruses, virus clearance is a major challenge for the immune system. These viruses can enter the nervous system using four pathways; (1) invasion of the sensory nerve endings, (2) invasion of the motor neurons at neuromuscular junctions, (3) invasion via the olfactory epithelium and olfactory neurons, and (4) invasion via infected circulatory leukocytes.

Neuro-invasiveness of SARS-Cov1 has been shown [15], and due to the high similarity between SARS-Cov1 and COVID-19 [16], COVID-19 can be suggested to invade the CNS. Previously in 2003, SARS-CoV was detected in the CSF of a SARS patient [17]. In addition, COVID-19 was detected in the CSF of a patient of meningitis/encephalitis associated with COVID-19 [18]. Covid-19 virus cell entry depends on ACE2 [19], which has been shown to be expressed in neurons [20]. Moreover, COVID-19 has been shown to infect neurons of human brain organoids [21,22], and COVID-19 was detected in the cortical neurons in brain autopsies of COVID-19 patients [22].

Interestingly, in our SAH case report as well as other reported COVID-19-positive SAH cases, the COVID-19 viral genome was not detected in the CSF, while the COVID-19 PCR test was positive for oropharyngeal swabs [10,11,12]. However, COVID-19 was detected in the CSF but not in the nasopharyngeal swab in a patient of meningitis/encephalitis associated with COVID-19 [18].

### 4.2. Endothelial Dysfunction

COVID-19 virus cell entry depends on the ACE2 receptor [19]. In addition to ACE2 [20], three other suggested mediators for COVID-19 entry into the cells, including extracellular matrix metalloproteinase inducer (CD147) [23], sialic acid [24], and transmembrane serine protease 2 [25] are present in endothelial cells and arterial smooth muscle cells [26]. The COVID-19 virus has been shown to infect and kill endothelial cells [27], which can destroy the inner layer of blood vessels and cause endothelial dysfunction. In addition, inflammatory cytokines such as IL-1 beta and TNF-alpha released by lymphocytes and macrophages can also cause endothelial dysfunction [28]. This results in the imbalance of factors released from endothelial cells including prostaglandin, nitric oxide, reactive oxygen species, and angiotensin, which can result in vasoconstriction [29], leading to vascular damage, which may lead to brain haemorrhage.

### 4.3. Inflammation

In COVID-19-positive patients, increased systemic inflammation through a disbalance of T-helper cells with an exaggerated Th1 response has been reported [30]. Similar kinds of alterations in T helper cell populations have been found in patients with intracranial aneurysms [31]. COVID-19 has been reported to induce cytokine storms with high levels of IL-1α, IL-1β IL-6, IL-8, IL-18, and TNF-α [32,33,34]. Moreover, the systemic immune inflammation index has been suggested as a potent marker for the worse clinical outcome of COVID-19 infected patients [35]. Systemic inflammation is known to cause vascular injury including the breakdown of collagen and the permeability of the blood–brain barrier [36,37], and a higher level of these inflammatory cytokines has been linked to the increased risk of intracranial aneurysm and aSAH [38,39]. In COVID-19 patients, a higher IL-6 level has been suggested as the predictor of poor outcome [40] and in a different study a higher level of IL-6 has been linked to poor outcome after aneurysmal rupture [41].

### 4.4. Case Description

A 61-year-old woman presented with dyspnea, cough, and fever. She was over-weight with a body mass-index of 34 and a history of hypertension. No history of subarachnoid hemorrhage in the family. She was admitted into ICU because of a low oxygen saturation (89%). A chest CT showed a typical picture of COVID-19 pneumonia (Figure 1A). A PCR test of an oropharyngeal swab was COVID-19-positive. She was prescribed with favipiravir and hydroxychloroquine in addition to non-invasive oxygen support. On the second day she experienced a sudden headache and a loss of consciousness. A subarachnoid haemorrhage in the basal cisterns from a ruptured anterior communicating artery aneurysm was revealed by computer tomography (CT) with CT-angiography (Figure 1B–G).

### 4.5. Treatment and Outcome

The aneurysm was clipped microsurgically through a standard pterional approach. All precautions including an FFP-3 mask, glasses for eye protection, and double gloves were used to protect surgical staff. The patient was admitted again to the intensive care unit for further intensive medical treatment. Post-operatively the patient showed slight motor dysphasia. No other neurological deficits.

## 5. Discussion

With increasing number of COVID-19 patients worldwide, multiple reports have shown that COVID-19 infection apart from the lungs can also involve other organs including the brain [6,8,42]. We present the case of a COVID-19 infection that led to an aneurysmal subarachnoid haemorrhage from a ruptured anterior communicating artery aneurysm. Microsurgical clipping was performed (due to lacking endovascular facility) to exclude the bleeding aneurysm from the circulation and pneumonia was treated with favipiravir and hydroxychloroquine in addition to oxygen support in an intensive medical ward.

Rupture of an intracranial aneurysm after COVID-19 infection might be a co-incidence. However, there are now multiple such reports in the literature. Case reports present the first line of evidence. Hence, here we collect the current evidence in the literature showing concomitant intracranial aneurysm ruptures and COVID-19 infections. The exact mechanism of aneurysm rupture after COVID-19 infection is unclear. There are however, multiple mechanisms that could explain how an intracranial aneurysm could possibly lead to vascular wall instability and aneurysm rupture. A severe cough in COVID-19 patients may cause extreme fluctuations in systemic blood pressure that may cause aneurysm rupture. COVID-19 positive patients with secondary lung damage need invasive ventilator support with high positive end expiratory pressure (PEEP) that may cause the elevation of systemic blood pressure and also intracranial pressure, which may contribute to aneurysm rupture.

Endothelial cells form the inner layer of the vascular wall. COVID-19-induced damage to endothelial cells could also be one of the possible mechanisms leading to aneurysm rupture. COVID-19 is associated with changes in endothelial morphology and apoptosis, which can accelerate the deterioration of the arterial wall leading to aneurysm rupture. Heat shock proteins (HSP) under shear and/or metabolic stress translocated from intracellular space to the plasma membrane and extracellular space can create the conditions to form autoimmunity. The COVID-19 proteins sharing the common epitope with heat shock proteins to elicit autoimmunity [43] when found on endothelial cells can cause endotheliitis. It can be suggested that already present shear stress on the walls of an aneurysm when associated with COVID-19 infection can trigger the translocation of these molecular chaperones to the endothelial cell membrane, leading to endotheliitis and vascular damage, which has been observed in various anatomical organs in COVID-19 patients [27]. ACE2 catalyzes the conversion of angiotensin 2 (Ang2) to Ang peptides. The decrease in ACE2 expression in endothelial cells caused by competitive binding and shedding induced by TNF-α and MMP-17 (ADAM17) [44,45] leads to the reduced conversion of Ang2 to Ang. This causes the accumulation of Ang2 [46], which is supported by the clinical reports showing higher Ang2 levels in COVID-19 patients [47]. High Ang2 concentration via endothelial cell death and disrupting the connections between endothelial cells and pericytes [48,49] results in vascular wall degeneration. In addition, Ang2, by causing vasoconstriction, increases the pressure on the already weakened aneurysm wall, which can result in aneurysm rupture. Furthermore, in the long term, COVID-19-induced endothelial cell damage along with increased inflammation can accelerate the growth of aneurysms, which can increase the probability of a rupture of a pre-COVID-19 stable aneurysm. Similar to intracranial aneurysm rupture, the sudden enlargement and rupture of aortic aneurysms has been reported, which has been attributed to endothelial damage and COVID-19-induced systemic inflammation and cytokine storms [50].

Another possible mechanism might be viral infection-triggered systemic inflammation. Viral infections including COVID-19 are known to induce cytokine storms (hypercytokinemia) leading to elevated systemic inflammation [51,52,53]. Systemic inflammation is known to cause vascular injury including the breakdown of collagen and permeability of the blood–brain barrier [36,37]. Influenza A virus infection for example disturbs the BBB through the involvement of systemic elevated MMP-9 [36], which breaks collagen present in the basal membrane of every arterial wall and a high collagen turnover in the systemic circulation is a sign of instability of existing intracranial aneurysm [54] in patients with unruptured intracranial aneurysms. Moreover, COVID-19 infection has been reported to increase systemic inflammation through a disbalance of T helper cells with an exaggerated Th1 response [30]. Similar kinds of alterations in T helper cell populations have been found in patients with intracranial aneurysms [31]. Moreover, a disturbed balance of macrophages and other inflammatory markers has been found in the wall of ruptured intracranial aneurysms [55,56] showing that inflammation is an important component of unstable aneurysms. In our patient, systemic leukocytes were elevated, which was a sign of systemic inflammation probably due to both infection and SAH. We could not, however, analyse subpopulations of different leukocytes in systemic circulation that might be altered. Another possibility could be a direct invasion of the virus in the brain as previously reported [42]. An aneurysm wall biopsy and COVID-19 PCR can exactly show the direct effect of the virus on the aneurysm wall. For the general practice, the neurosurgeons should work closely with other disciplines to rapidly diagnose and treat such patients.

Multiple deaths after COVID-19 infection are attributed to systemic organ failure and the cited cases in this study might be under reported. In combination with other case reports from the literature, we conclude that COVID-19 infection might increase the risk of aneurysm enlargement and rupture probably through extreme ventilation or cough-associated fluctuations in systemic blood pressure or involving endothelial cell damage along with a systemic inflammation-mediated mechanism. However, more epidemiological/clinical studies are requisite to confirm the relationship and animal experiments in controlled conditions are needed to find out the exact mechanism.

## 6. Conclusions

Systemic inflammation and ventilator support-associated blood pressure fluctuations may trigger aneurysmal subarachnoid haemorrhages secondary to COVID-19 infection. COVID-19 infection could be considered as one of the possible risk factors leading to the instability and rupture of intracranial aneurysms. We suggest that people with known intracranial aneurysms and who also had a COVID-19 infection may need more frequent surveillance of the aneurysm progress until we understand the relationship between COVID-19 infection and intracranial aneurysm pathophysiology.

## Figures and Tables

**Figure 1 pathogens-11-00617-f001:**
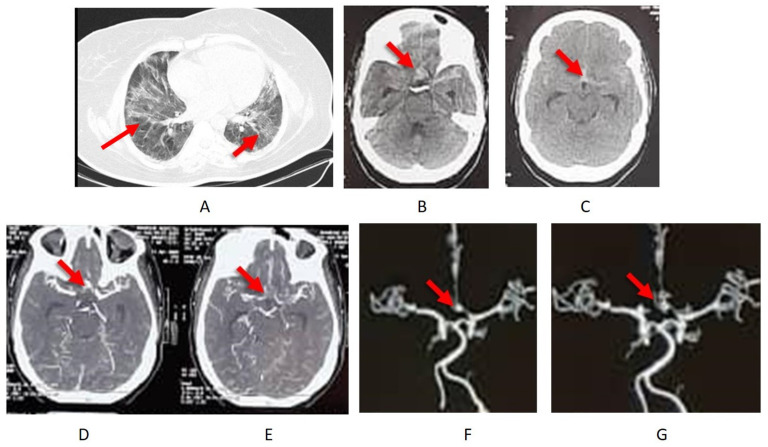
COVID-19 infection with pneumonia (**A**) and subarachnoid haemorrhage (**B**,**C**) from an anterior communicating artery aneurysm (**D**–**G**).

**Table 1 pathogens-11-00617-t001:** Clinical characteristics, presentation, and outcome of COVID-19-positive SAH patients.

Aneurysmal SAH	6 Cases
Age	57 ± 13
Gender	Male 16% and female 83%
Initial symptoms	Fever, fatigue, dry cough, shortness of breath, nausea, vomiting, anosmia, headache, dyspnea, myalgia, confusion, and loss of consciousness
Neurological symptoms	Anosmia, mental confusion, headache associated with right hemiparesis aphasia, seizures, loss of consciousness, headache, and vomiting
Comorbidies	Previously detected aneurysm (16%)
	Systemic hypertension (16%)
	Obesity and systemic hypertension (16%)
Hydrocephalus	33%
Respiratory support	Mechanical ventilation 54% and nasal oxygen therapy 33%
SAH treatment	Microsurgical clipping 50%, occluded with coils 16%, balloon-assisted embolization 16%, and treated with a flow-diverter 16%
EVD	33%
Outcomes	67% discharged for rehabilitation
	33% discharged asymptomatic

## Data Availability

Not applicable.

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
