# Peer review of "Intracranial Aneurysm Rupture after SARS-CoV2 Infection: Case Report and Review of Literature"

_pathogens, 2022, doi:10.3390/pathogens11060617_

Round 1

Reviewer 1 Report

In this article, the authors discuss the relationship between COVID-19 infection and subarachnoid haemorrhage (SAH) based on an own case and cases available in the literature. The article can be published, but still requires some minor corrections:

Please update your details concerning COVID-19 worldwide – in the first sentence: “The number of SARS-CoV-2 virus infected patients is increasing dramatically and have now reached above 100 million confirmed infected individuals with over 2.3 million deaths worldwide.”

Correct in Table 1: “pystemic hypertension”

Text editing needs improvement (correct font, correct punctuation and capitalization errors).

Author Response

Dear Reviewer,

We are thankful for the positive constructive comments. We have corrected the details from the WHO latest data, according to which there are above 520 million confirmed infected individuals with over 6.2 deaths worldwide (page 1, line 38). We have also corrected the spelling error in the table.

We have improvised the manuscript. We did not change the font because we think that this will be automatically done by the journal.

Reviewer 2 Report

The case report entitled “Intracranial Aneurysm Rupture after SARS-CoV2 Infection: 2 Case Report and Review of Literature” by Dilaware Khan et al.  is certainly a remarkable testimony to the fact that COVID 19 is not just an airway disease but a systemic disease.

In the discussion, the authors are assessed several possible causes that may be involved in vascular instability during COVID 19. A metabolic process that could be the basis of various endothelial manifestations due to the inflammatory processes that are triggered in COVID, is certainly molecular mimicry. In the literature there are several works on the role of molecular mimicry and on the involvement of stress proteins, Hsp, in inflammatory processes involving the endothelium of various anatomical organs. I therefore suggest that the authors include these aspects in the discussion as well.

Author Response

We are thankful for the constructive comments and suggestions to improve the manuscript. In the discussion, we have added endotheliitis and vascular damage resulting from shear stress associated COVID-19 induced translocation of HSPs to the endothelial membrane (Page 6, line 191-198). 

Reviewer 3 Report

Thank you for submitting your interesting manuscript. I find it well-structured and well-written. However, I suggest that you improve the following aspects before it can be considered for publication:

  • Please, specify the research method more in detail: which databases and which keywords did you use and how many results did you exclude for not matching the inclusion criteria?
  • In describing your own case, you did not specify which type of oxygen support the patient was receiving, which is, according to your statements, a relevant information. You should also comment on the fact that, in the review, most patients were receiving mechanical ventilation: do you think it might be at higher risk of aneurysm rupture than NIV?
  • Figg 1F and 1G quality is very poor, is it possible to have a better resolution?
  • The hypothesis that Covid infection could be responsible for sudden aneurysm rupture is fascinating, although it cannot be proved based on your results. In order to strengthen it, you could look for ruptured non-cerebral aneurysms (e.g. aortic) in Covid-patients and cite some relevant works reporting on that aspect.
  • The grammar of some sentences should be checked (lines 207-8, for instance, should be rephrased: what is the above mentioned??)

Author Response

We are thankful for the constructive criticism. Following the suggestions, we have improvised the manuscript.

  • We searched for the clinical reports using the keyword ‚Subarachnoid Haemorrhage (SAH)‘ in PubMed and on google scholar. We have added this information in the method section We excluded the case reports with non-aneurysmal subarachnoid haemorrhage and with aetiology unknown.

  • The patient was provided with noninvasive oxygen support without mechanical ventilation. Mechanical ventilation could potentially contribute to aneurysm rupture. In the discussion, we have stated the possibility that invasive ventilation support with high positive end-expiratory pressure (PEEP) may cause elevation of systemic blood pressure and also intracranial pressure that may consequently contribute to aneurysm rupture.

  • We apologize that unfortunately, we had only these figures available.

  • Following the reviewer,s suggestions we have included this information (page 6, line 209-212).

  • We have revised the manuscript and improvised it.

Round 2

Reviewer 3 Report

Most comments have been addressed. I recommend acceptance of the manuscript

Author Response

We thank the reviewer for his/her valuable comments.